# Reduced-Complexity Multiple-Symbol Detection of O-QPSK Signals in Smart Metering Utility Networks

**Congyu Shi [1]**, **Gaoyuan Zhang [1,2,3,*]**, **Haiqiong Li [1]**, **Congzheng Han [2]**, **Jie Tang [3]**,
**Hong Wen [3]**, **Longye Wang [3,4] and Dan Wang [1]**

[1] School of Information Engineering, Henan University of Science and Technology, Luoyang 471023, China; cyshi1996@163.com (C.S.); lihaiqiong96@163.com (H.L.); wangdaniel2004@163.com (D.W.)

[2] Key Laboratory of Middle Atmosphere and Global Environment Observation, Institute of Atmospheric Physics, Chinese Academy of Sciences, Beijing 100029, China; c.han@mail.iap.ac.cn

[3] National Key Laboratory of Science and Technology on Communications, University of Electronic Science and Technology of China, Chengdu 611731, China; cs.tan@163.com (J.T.); sunlike@uestc.edu.cn (H.W.); 201311260116@std.uestc.edu.cn (L.W.)

[4] School of Engineering and Technology, Tibet University, Lhasa 850000, China

* Correspondence: zhanggaoyuan407@163.com; Tel.: +86-379-6423-1910

**Abstract:** In this work, an implementation-friendly multiple-symbol detection (MSD) scheme is proposed for the IEEE 802.15.4g offset quadrature phase shift keying (O-QPSK) receivers over the slow fading channel. The full MSD scheme presents better detection performance than the symbol-by-symbol detection (SBSD) scheme, yet its complexity increases exponentially as the observation window length increases. We introduce a simplified MSD scheme based on two powerful strategies. We first seek the optimal and suboptimal decisions in each symbol position with the standard SBSD procedure. Then, the aforementioned optimal and suboptimal decisions instead of all candidates are jointly searched with the standard MSD procedure. That is, only the most and second most reliable candidates in each symbol position are selected to participate in the final detection. The simulation results demonstrate that the new MSD scheme can achieve more encouraging energy gain than the SBSD scheme, while the high complexity of full MSD is also effectively reduced. A more legitimate compromise between detection performance and complexity is thus accomplished, which enables smart metering utility networks (SUN) nodes to achieve energy saving and maximum service life.

**Keywords:** smart metering utility networks; IEEE 802.15.4g; offset QPSK; multiple-symbol detection

## 1. Introduction

In recent years, pervasive wireless sensor networks (WSNs) have received much concentration, especially due to their extensive application potential in 5G-enabled Internet of things (IoT), whose typical scenarios include smart metering utility networks (SUN), industrial wireless control, environmental monitoring and so on [1–3]. How to reliably and efficiently transmit the sensed data to the receiver within one hop is of great significance for distributed WSNs nodes [4,5]. IEEE 802.15.4g standard is tailored for ultra-low-power wireless communication systems in SUN, wherein the low-cost offset quadrature phase shift keying (O-QPSK) physical layer has aroused much interest of scholars [6,7]. Note that, although there are few researches on the detection mechanism of receivers, such researches are equally important for scholars and users. This is mainly because the detection performance of receivers can directly affect the signal recovery efficiency of receivers and the transmission efficiency of WSNs nodes, and even affect the working efficiency of the entire WSNs [8].

In contrast to previous studies which are mainly focused on symbol-by-symbol detection (SBSD) [9,10], we turn our attention to the method of increasing observation window length for further reliability improvement of IEEE 802.15.4g O-QPSK receivers, which has a significant impact on the improvement of the WSNs nodes' reliability [11–14]. That is, the multiple-symbol detection (MSD) scheme is investigated. Note that the MSD scheme exhibits better performance, yet its high complexity brings considerable challenges to the detection process.

The MSD scheme based on maximum likelihood (ML) can obtain the optimal detection performance, but its complexity will increase exponentially as the observation window length increases [15–17]. Therefore, it is significant to decrease the complexity of MSD scheme. In recent years, scholars have proposed various algorithms to solve this problem [18–29]. The main related work is as follows. Stephen G. Wilson et al. introduced the strategy of optimal block detection to reduce the complexity of MSD scheme, while its block length increased exponentially [21]. Kenneth M. et al. introduced a fast-implemented MSD algorithm in [22], and Li Bin considered decreasing the search factor of detection procedure in [23,24]. Lutz Lampe et al. applied spherical decoding to ML MSD in time-varying Rayleigh fading channels in [25,26]. Reference [27] derived a frequency-insensitive MSD strategy that approximates ML. Furthermore, G.Y. Zhang et al. proposed a novel and low-complexity MSD scheme with the aid of preamble, which is verified to be valid in the IEEE 802.15.4 BPSK receivers [28–30]. Thus, it is imperative to develop an implementation-friendly MSD scheme for the IEEE 802.15.4g O-QPSK receiver.

In this work, we propose a simplified MSD scheme for IEEE 802.15.4g O-QPSK receivers. The high complexity of the full MSD scheme is reduced while most reliability is also preserved. The main contributions of this work are as follows:

- The MSD scheme based on the maximum likelihood criterion can acquire impressive performance, yet it is hard to implement. Following a heuristics configuration, we introduce an implementable full MSD scheme, which meets the performance requirements of the IEEE 802.15.4 standard [6,7].
- We introduce a novel and simplified MSD scheme of the O-QPSK receiver over pure additive white Gaussian noise (AWGN) and slow fading channel. In particular, when the observation window length is set to be 2 and only the maximum and submaximum metrics are considered, our enhanced detector presents acceptable performance.
- A carrier frequency offset (CFO) estimator that matches the proposed simplified MSD scheme is found based on the previous researches of Michael P. Fitz [20]. Specially, we simplified the CFO estimator following from our previous researches.
- The detection characteristics of our proposed MSD scheme are investigated from diverse aspects through experimental simulation. To examine the robustness of the proposed MSD scheme to carrier phase offset (CPO), we also specially investigated the performance evaluation results of the proposed detection scheme under the condition of dynamic CPO.
- In order to illustrate the energy-saving characteristics of our proposed algorithm for transmission-only nodes, we also analyzed the transmitter energy consumption in a real sensor node platform, namely, Atmel AT86RF215 [31,32], and compared the energy consumption gain of the proposed MSD scheme configured various estimators.

The remainder of this paper is distributed as follows. Section 2 introduces the signal model over a slow fading channel. In Section 3, the full MSD scheme is described. Section 4 concentrates on the detailed process of our proposed simplified MSD scheme. Section 5 is the evaluated carrier frequency offset effect (CFOE) estimation scheme. Section 6 offers the numerical results and discussions, and Section 7 indicates the conclusions and future research directions.

## 2. Signal Model

In this work, we consider a slow fading channel with perfect synchronization [5]. The received signal can be described as follows

$$r(t) = h(t)s(t)e^{j(2\pi ft + \theta)} + n(t) \tag{1}$$

Here, $s(t)$ denotes the transmitted chip baseband signal, and $h(t)$ is the multiplicative fading. $f$ and $\theta$ are the CFO and CPO, respectively. $n(t)$ represents an AWGN with a double sideband power spectral density $N_0/2$. In particular, $s(t)$ can be expressed as

$$s(t) = \sum_{k=-\infty}^{\infty} s_k p(t - kT - \tau) \tag{2}$$

where $p(t)$ denotes the pulse shape, and $s_k$ is the modulation symbols. $\tau$ is the channel delay, $T$ is the symbol period and the sampling interval is $kT + \tau$.

The continuous signal $r(t)$ can be converted into discrete sequence $r_{m,k}$ by the filter matching to $p(t)$ [5]. Especially, assuming perfect carrier synchronization and no inter-symbol interference, the discrete-time complex baseband equivalent received chip sequence of the $m$th symbol $E[m]$ is as follows.

$$r_{m,k} = h_{m,k}s_{m,k}e^{j(2\pi f_{m,k}kT_c + \theta_{m,k})} + n_{m,k}, \quad 1 \le k \le K/2 \tag{3}$$

Here, $h_{m,k}$ is multiplicative fading. $s_{m,k}$ denotes the $k$th complex chip in the $m$th symbol interval, $s_{m,k} \in \{\pm 1 \pm j\}$. $f_{m,k}$ and $\theta_{m,k}$ represent the CFO and CPO in radians, respectively. $T_c$ denotes the chip period, and $n_{m,k}$ is the samples taken from AWGN $n(t)$. $K = 32$ is the length of the pseudo-random noise (PN) sequence.

Assuming that CFO and CPO are random and unknown at receiver, $h_{m,k}$, $f_{m,k}$, $\theta_{m,k}$ and $n_{m,k}$ are independent of each other. Specially, this work pay attention to the case where $h_{m,k} = h$, $f_{m,k} = f$, $\theta_{m,k} = \theta$, $n_{m,k} = n$ across a packet transmission.

## 3. General Full Multiple-Symbol Detection Scheme

The optimal MSD scheme can be obtained based on the ML criterion, yet its high complexity is unfavorable to design low-power, low-cost WSNs nodes [15]. In this work, we consider a heuristic idea. The CFOE is first estimated and compensated based on the preamble, and then the full MSD scheme only with unknown CPO is configured [5]. The following is the specific steps of the detection process.

First, the CFO $r_{m,k}$ is estimated by

$$r'_{m,k} = r_{m,k}e^{-jk\hat{\varphi}} \tag{4}$$

Here, $\hat{\varphi} \triangleq \hat{\omega}T_c = 2\pi\hat{f}T_c$, and $\hat{f}$ is the estimator of $f$. The estimation of CFO is extremely important, which will be described in detail in Section 5. Note that we assume that the CFO is perfectly estimated; that is, the effect of nuisance parameter $f$ of $r_{m,k}$ has been completely eliminated after compensation.

Then, the whole symbol sequence is divided into blocks, and each block contains $j$ symbols. The decision statistic of the $i$th observation window can be expressed as

$$Y_{i_x} = \left| \sum_{m=j(i-1)+1}^{ij} \sum_{k=1}^{M} r'_{m,k}s^*_{p,k} \right|^2, 1 \le i_x \le 16^j \tag{5}$$

Here, $*$ represents the complex conjugate operation, $s_{p,k}$ is the $k$th chip in the $p$th PN sequence $s_p$ and $1 \leq p \leq 16$. $M$ is the truncated number of chips and $1 \leq M \leq K/2$. Note that it is the SBSD scheme when $j$ is set to be 1 [8].

Next, the decision rule of the $i$th observation window can be given by

$$\hat{Y}[i] = \underset{1 \leq i_x \leq 16^j}{\arg \max} \left\{ Y_{i_x} \right\} \tag{6}$$

Finally, according to the optimal decision $\hat{Y}[i]$, the output bit detection sequences $\left\{ \hat{E}[m], j(i-1) + 1 \leq m \leq ij \right\}$ of the $i$th observation window can be obtained.

For full MSD scheme, even setting the observation window length $j$ to be 2, the decision statistic $Y_{i_x}$ still inevitably needs to be calculated 256 times. This high complexity is undesirable. To effectively decrease the implementation complexity, this paper introduces a simplified MSD scheme in Section 4. Section 6 quantitatively describes the proposed MSD scheme through experimental simulation.

## 4. The Proposed Multiple-Symbol Detection Scheme

In our proposed simplified MSD scheme, the local decision metric $Y_{i_x}$ in each observation window is first searched for its optimal and suboptimal decision, and then the maximum is determined for the $2^j$ local decision metrics in each observation window. Specifically, we take the observation windows of 3 as an example to describe this scheme, which is similar when selecting other observation windows length. The following are the detailed detection steps.

First, for the $i$th observation window, the decision metric of each symbol can be expressed as

$$Z_{3(i-1)+l,p} = \left| \sum_{k=1}^{M} r'_{3(i-1)+l,k} s^*_{p,k} \right|^2, \quad i \geq 1, 1 \leq l \leq 3, 1 \leq p \leq 16 \tag{7}$$

where $M$ is set to be the maximum of 16. $r'$ is the compensated complex baseband sample, which can be obtained by (4), and $s_{p,k}$ is the PN sequence.

Next, looking for the maximum and the submaximum decision of the metric $Z_{3(i-1)+l,p}$, we can get the local maximum and submaximum statistic of the $j$th symbol in the $i$th observation window as follows:

$$Z_{3(i-1)+l,a_{3(i-1)+l}} = \underset{1 \leq p \leq 16}{\arg \max} \left\{ Z_{3(i-1)+l,p} \right\}, \quad 1 \leq l \leq 3, a_{3(i-1)+l} \in \{p \,|\, 1 \leq p \leq 16\} \tag{8}$$

$$Z_{3(i-1)+l,b_{3(i-1)+l}} = \underset{1 \leq p \leq 16, b_{3(i-1)+l} \neq a_{3(i-1)+l}}{\arg \max} \left\{ Z_{3(i-1)+l,p} \right\},$$
$$1 \leq l \leq 3, b_{3(i-1)+l} \in \left\{ p \,|\, 1 \leq p \leq 16, b_{3(i-1)+l} \neq a_{3(i-1)+l} \right\} \tag{9}$$

Here, $a_l$ and $b_l$ respectively represent the maximum and the submaximum subscript of the decision metric $Z_{3(i-1)+l,p}$ of the $l$th symbol in the $i$th observation window, and $\boldsymbol{a} = \left\{ a_{3(i-1)+l} \right\}$, $\boldsymbol{b} = \left\{ b_{3(i-1)+l} \right\}$.

For the $i$th observation window containing 3 symbols, 3 local maximum statistics $Z_{3(i-1)+l,a_l}$ and 3 local submaximum statistics $Z_{3(i-1)+l,b_l}$ can be obtained; that is, a total of 8 results. In particular, for the full MSD scheme, the decision statistics contain 4096 values. It can be seen that the simplified MSD scheme reduces the decision statistics by a factor of 512, which not only dramatically reduces the computational complexity of the detection process but is also expected in the design of low-power, low-cost WSNs nodes.

Then, these 8 local decision statistics are judged again, and the decision metric $Y_{i_x}$ can be obtained,

$$Y_{i_x} = \left| \sum_{m=3(i-1)+1}^{3i} \sum_{k=1}^{16} r'_{m,k} s^*_{\hat{p}_l,k} \right|^2, \quad i \geq 1, 1 \leq l \leq 3, 1 \leq i_x \leq 8 \tag{10}$$

where the 3 PN sequences corresponding to the decision metric $Y_{i_x}$ of the $i$th observation window are set to be $s_{i_x} = \{s_{\hat{p}_1,k}, s_{\hat{p}_2,k}, s_{\hat{p}_3,k}\}$, $\hat{p}_m \in \{a(m), b(m)\}$.

Next, find the maximum of the decision metric $Y_{i_x}$, and freeze the corresponding $s_{i_x}$, which is denoted as $s_{i_x} = \{\hat{s}_{\hat{p}_1,k}, \hat{s}_{\hat{p}_2,k}, \hat{s}_{\hat{p}_3,k}\}$. We can acquire an equivalent but low-complexity decision metric $Y_{i_x}$.

$$\hat{Y}[i] = \arg\max_{1 \leq x \leq 2^3 = 8} \{Y_{i_x}\} \tag{11}$$

Finally, the output bit information $\{\hat{E}[m]\}$ can be acquired by performing demapping according to the decision metric $\hat{Y}[i]$. The receiver structure of our proposed MSD scheme is shown in Figure 1. Algorithm 1 summarizes the specific implementation process of the simplified MSD algorithm. Note that, for the sake of presentation, we only select the local maximum and the local submaximum decision of the statistic $Z_{3(i-1)+l,p}$, and the parameters in Figure 1 and Algorithm 1 are same as this section. Essentially, when the number of metric factor $p$ increases to 16, it corresponds to the general full MSD scheme. In particular, in Section 6, we not only analyzed the performance of simplified two-symbol detection and three-symbol detection but also increased $p$ to 4 for quantitative simulation analysis.

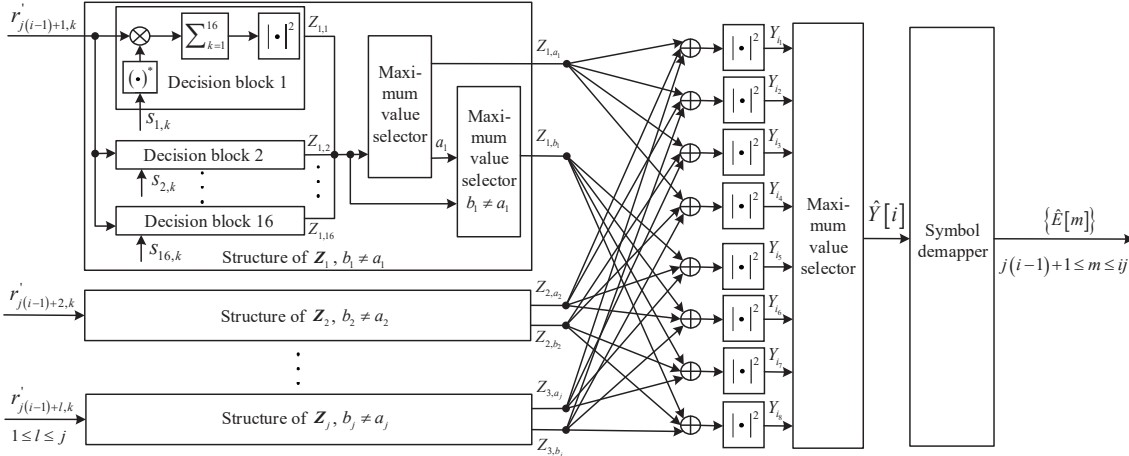

**Figure 1.** The receiver structure of our proposed multiple-symbol detection (MSD) scheme.

---

**Algorithm 1** The proposed multiple-symbol detection algorithm.

---

**Input:** $\hat{\varphi}$: detection of the actual information phase;

　　$r_{m,k}$: complex baseband samples of the $m$th symbol period $E[m]$;

　　$s_{p,k}$: complex baseband pseudo-random noise (PN) sequence in the $p$th symbol interval, and $1 \leq$
　　$p \leq 16$;

　　$L$: sample symbol number of the actual data;

　　$L_1$: payload length of the physical layer protocol data unit (PPDU);

　　$N$: the number of truncated differential chips, $1 \leq N \leq 5$, and here we set $N = 3$;

　　$J$: the maximum length of preamble, and $J = 8$;

　　$K$: chip length of the PN sequence, and $K = 32$;

　　$M$: sample chip number for each symbol of the actual data, and $1 \leq M \leq K/2$. For better result
　　presentation, set the maximum value of $M = 16$.

**Output:** $\{\hat{E}[m]\}$: Detection of $j$ actual data symbols in the $i$th observation window, and $j(i-1)+1 \leq$
　　$m \leq ij$.

1: Initialize $L_1 = 176$, $j = 3$, $M = 16$;
2: **for** $i = 1$; $i \leq [(L+J)/j]$; $i++$ **do**
3: 　　**for** $m = j(i-1)+1$; $m \leq ij$; $m++$ **do**
4: 　　　　**for** $p = 1$; $p \leq 16$; $p++$ **do**
5: 　　　　　　**for** $k = 1$; $k \leq M$; $k++$ **do**
6: 　　　　　　　　$r'_{m,k} \leftarrow r_{m,k}e^{-jk\hat{\varphi}}$;
7: 　　　　　　　　$Z_{m,p} \leftarrow Z_{m,p} + r'_{m,k}s^*_{p,k}$;
8: 　　　　　　**end for**
9: 　　　　**end for**
10: 　　**end for**
11: 　　$Z_{m,p} \leftarrow |Z_{m,p}|^2$;
12: **end for**
13: **for** $m = j(i-1)+1$; $m = j(i-1)+1$; $m++$ **do**
14: 　　$\hat{Z}_{m,a_m} \leftarrow \underset{1 \leq p \leq 16}{\arg\max} \{Z_{m,p}\}$;
15: 　　$\hat{Z}_{m,b_m} \leftarrow \underset{1 \leq p \leq 16, b_m \neq a_m}{\arg\max} \{Z_{m,p}\}$;
16: **end for**
17: Save the maximum and sub-maximum indices of the measure $\mathbf{Z}_m$, and $\mathbf{a} = a_m$, $\mathbf{b} = b_m$.
18: **for** $m = j(i-1)+1$; $m \leq ij$; $m++$ **do**
19: 　　**for** $k = 1$; $k \leq M$; $k++$ **do**
20: 　　　　$Y_{i_x} \leftarrow Y_{i_x} + r'_{m,k}s^*_{\hat{p}_m,k}$;
　　Notice that, $1 \leq i_x \leq 2^j$, and here $\hat{p}_m \in \{\mathbf{a}(m), \mathbf{b}(m)\}$.
21: 　　**end for**
22: **end for**
23: $Y_{i_x} \leftarrow |Y_{i_x}|^2$;
24: **for** $i_x = 1$; $i_x \leq 2^j$; $i_x++$ **do**
25: 　　$\hat{Y}[i] \leftarrow \underset{1 \leq i_x \leq 2^j}{\arg\max} \{Y_{i_x}\}$, and freeze the corresponding $s_{\hat{p}_m,k}$;
26: **end for**
27: Obtain detection information $\{\hat{E}[m]\}$ in the $i$th Observation window by the decision measure $\hat{Y}[i]$
　　and $s_{\hat{p}_m,k}$.
28: **return** $\{E[m]\}$.

---

## 5. Proposed Estimation Scheme

The issue of CFOE estimation has always been a concern in signal processing. Optimal maximum likelihood estimation (MLE) is a well-known strategy, which can approach the Cramer–Raw limit under the condition of sufficiently high SNR [15]. However, in many cases, even if fast Fourier transform (FFT) algorithm is employed, the calculation results are prohibitive [5], so a more straightforward and higher-efficiency CFO estimation algorithm is urgently needed. In this work, based on a heuristic idea, we proposed a simplified CFOE estimation algorithm inspired by the previous work of Michael P. Fitz [20] and G.Y. Zhang et al. [29]. The detailed algorithm structure is shown in Figure 2, whose particular procedure is as follows.

First, by preprocessing the chip samples, we can get a differential autocorrelation function $R(n)$ as follows.

$$R(n) = \frac{1}{J(M-n)} \sum_{m=1}^{J} \sum_{k=n+1}^{M} x_{m,k} x^*_{m,k-n}$$
$$= |h|^2 e^{2\pi f n T_c} + \lambda, \ 1 \leq n \leq Q, 2 \leq k \leq M \tag{12}$$

Here, $x_{m,k}$ is given by

$$x_{m,k} = r_{m,k} s^*_{m,k}$$
$$= h e^{j(2\pi f k T_c + \theta)} + \lambda' \ 1 \leq k \leq K/2 \tag{13}$$

where $\lambda$ and $\lambda'$ are the comprehensive noise term. $J$ denotes the number of preambles, and we consider the maximum of it here, that is, $J = 8$. $n$ is the number of chip delay, and $1 \leq n \leq Q$, which makes the real phase in the principal value period. $Q = 5$ is the available maximum of chip delay [20].

Next, following the previous work of Michael P. Fitz et al., the quantization function of the CFOE can be expressed as

$$G(N) \approx \frac{6}{N(N+1)(2N+1)} \sum_{n=1}^{N} n \arg\{R(n)\}$$
$$= \frac{\sum_{n=1}^{N} n \arg\{R(n)\}}{\sum_{n=1}^{N} n^2} \tag{14}$$
$$= \sum_{n=1}^{N} C(N,n) \arg\{R(n)\}, \quad N \leq Q$$

Here, $\arg\{\bullet\}$ denotes the argument of a complex number. $N$ is the truncated number, and $N \leq Q$. $C(N,n)$ is the delay weighting factor of $R(n)$.

Then, the CFOE can be visually estimated as

$$\hat{\varphi} \stackrel{\Delta}{=} \hat{\omega} T_c = 2\pi \hat{f} T_c = G(N) \tag{15}$$

In this paper, the estimation of the residual CFO is based on the detection of the signal envelope. Compared with the common space subdivision method, the adaptive CFO estimation algorithm with the triangular approximation algorithm provides a simpler space division rules and without undesirable errors [8,9]. Moreover, the complexity of the full estimator can achieve significant reduction by the triangular approximation $\tan^{-1}(x) = x$ and $\sin^{-1}(x) = x$. We can easily achieve two full but high-complexity complex expressions of $\arg\{R(n)\}$ as shown in (16) and (17) [8].

$$
\arg\{R(n)\} =
\begin{cases}
\tan^{-1}\dfrac{\text{Im}\,[R(n)]}{\text{Re}\,[R(n)]}, & \text{if } \text{Re}\,[R(n)] > 0 \text{ and } |\text{Re}\,[R(n)]| \geq |\text{Im}\,[R(n)]| \\[2ex]
\dfrac{\pi}{2} - \tan^{-1}\dfrac{\text{Re}\,[R(n)]}{\text{Im}\,[R(n)]}, & \text{if } \text{Im}\,[R(n)] > 0 \text{ and } |\text{Re}\,[R(n)]| < |\text{Im}\,[R(n)]| \\[2ex]
-\pi + \tan^{-1}\dfrac{\text{Im}\,[R(n)]}{\text{Re}\,[R(n)]}, & \text{if } \text{Re}\,[R(n)] < 0 \text{ and } |\text{Re}\,[R(n)]| \geq |\text{Im}\,[R(n)]| \\[2ex]
-\dfrac{\pi}{2} - \tan^{-1}\dfrac{\text{Re}\,[R(n)]}{\text{Im}\,[R(n)]}, & \text{if } \text{Im}\,[R(n)] < 0 \text{ and } |\text{Re}\,[R(n)]| < |\text{Im}\,[R(n)]|
\end{cases}
\tag{16}
$$

$$
\arg\{R(n)\} =
\begin{cases}
\sin^{-1}\dfrac{\text{Im}\,[R(n)]}{\sqrt{\text{Re}^2\,[R(n)] + \text{Im}^2\,[R(n)]}}, \\[1ex]
\qquad \text{if } \text{Re}\,[R(n)] > 0 \text{ and } |\text{Re}\,[R(n)]| \geq |\text{Im}\,[R(n)]| \\[2ex]
\dfrac{\pi}{2} - \sin^{-1}\dfrac{\text{Re}\,[R(n)]}{\sqrt{\text{Re}^2\,[R(n)] + \text{Im}^2\,[R(n)]}}, \\[1ex]
\qquad \text{if } \text{Im}\,[R(n)] > 0 \text{ and } |\text{Re}\,[R(n)]| < |\text{Im}\,[R(n)]| \\[2ex]
-\pi - \sin^{-1}\dfrac{\text{Im}\,[R(n)]}{\sqrt{\text{Re}^2\,[R(n)] + \text{Im}^2\,[R(n)]}}, \\[1ex]
\qquad \text{if } \text{Re}\,[R(n)] < 0 \text{ and } |\text{Re}\,[R(n)]| \geq |\text{Im}\,[R(n)]| \\[2ex]
-\dfrac{\pi}{2} + \sin^{-1}\dfrac{\text{Re}\,[R(n)]}{\sqrt{\text{Re}^2\,[R(n)] + \text{Im}^2\,[R(n)]}}, \\[1ex]
\qquad \text{if } \text{Im}\,[R(n)] < 0 \text{ and } |\text{Re}\,[R(n)]| < |\text{Im}\,[R(n)]|
\end{cases}
\tag{17}
$$

Considering the triangular approximation algorithm $\tan^{-1}(x) \approx x$ and $\sin^{-1}(x) \approx x$ in our previous work [8], we can obtain two simplified expressions of $R(n)$ as given by Equations (18) and (19).

$$
\arg\{R(n)\} \approx
\begin{cases}
\dfrac{\text{Im}\,[R(n)]}{\text{Re}\,[R(n)]}, & \text{if } \text{Re}\,[R(n)] > 0 \text{ and } |\text{Re}\,[R(n)]| \geq |\text{Im}\,[R(n)]| \\[2ex]
\dfrac{\pi}{2} - \dfrac{\text{Re}\,[R(n)]}{\text{Im}\,[R(n)]}, & \text{if } \text{Im}\,[R(n)] > 0 \text{ and } |\text{Re}\,[R(n)]| < |\text{Im}\,[R(n)]| \\[2ex]
-\pi + \dfrac{\text{Im}\,[R(n)]}{\text{Re}\,[R(n)]}, & \text{if } \text{Re}\,[R(n)] < 0 \text{ and } |\text{Re}\,[R(n)]| \geq |\text{Im}\,[R(n)]| \\[2ex]
-\dfrac{\pi}{2} - \dfrac{\text{Re}\,[R(n)]}{\text{Im}\,[R(n)]}, & \text{if } \text{Im}\,[R(n)] < 0 \text{ and } |\text{Re}\,[R(n)]| < |\text{Im}\,[R(n)]|
\end{cases}
\tag{18}
$$

$$
\arg\{R(n)\} \approx
\begin{cases}
\dfrac{\text{Im}\,[R(n)]}{\sqrt{\text{Re}^2\,[R(n)] + \text{Im}^2\,[R(n)]}}, & \text{if } \text{Re}\,[R(n)] > 0 \text{ and } |\text{Re}\,[R(n)]| \geq |\text{Im}\,[R(n)]| \\[2ex]
\dfrac{\pi}{2} - \dfrac{\text{Re}\,[R(n)]}{\sqrt{\text{Re}^2\,[R(n)] + \text{Im}^2\,[R(n)]}}, & \text{if } \text{Im}\,[R(n)] > 0 \text{ and } |\text{Re}\,[R(n)]| < |\text{Im}\,[R(n)]| \\[2ex]
-\pi - \dfrac{\text{Im}\,[R(n)]}{\sqrt{\text{Re}^2\,[R(n)] + \text{Im}^2\,[R(n)]}}, & \text{if } \text{Re}\,[R(n)] < 0 \text{ and } |\text{Re}\,[R(n)]| \geq |\text{Im}\,[R(n)]| \\[2ex]
-\dfrac{\pi}{2} + \dfrac{\text{Re}\,[R(n)]}{\sqrt{\text{Re}^2\,[R(n)] + \text{Im}^2\,[R(n)]}}, & \text{if } \text{Im}\,[R(n)] < 0 \text{ and } |\text{Re}\,[R(n)]| < |\text{Im}\,[R(n)]|
\end{cases}
\tag{19}
$$

In order to present our simplified CFOE estimator more clearly, we describe the calculation process of the CFOE quantization function $G(N)$ in detail in Algorithm 2, and give its specific structure in Figure 2.

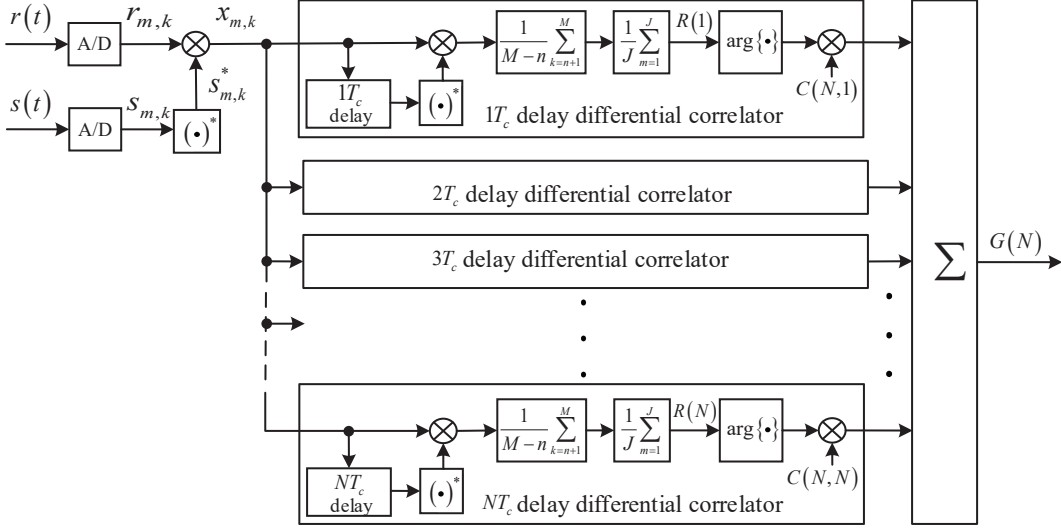

**Figure 2.** The structure of the quantization function $G(N)$.

---

**Algorithm 2** The proposed CFO estimation algorithm.

---

**Input:** $r_{m,k}$: complex baseband samples of the $m$th symbol period $E[m]$;

    $s_{m,k}$: the $k$th complex baseband O-QPSK modulation chip in the $m$th symbol interval $E[m]$;

    $N$: the number of truncated differential chips, $1 \leq N \leq 5$, and here we set $N = 3$;

    $J$: the maximum length of preamble, and $J = 8$;

    $K$: chip length of the PN sequence, and $K = 32$;

    $M$: sample chip number for each symbol of the actual data and $1 \leq M \leq K/2$. For better result

    presentation, set the maximum value of $M = 16$.

**Output:** $\hat{\varphi}$: Detection of the actual information phase.

1: Initialize $N = 3$, $J = 8$, $1 \leq M \leq K/2$, $K = 32$, and $M = 16$;
2: **for** $n = 1, n \leq N, n + +$ **do**
3:     **for** $m = 1, m \leq J, m + +$ **do**
4:         **for** $k = 1, k \leq M, k + +$ **do**
5:             $x_{m,k} \leftarrow r_{m,k} s_{m,k}^*$;
6:         **end for**
7:     **end for**
8:     **for** $m = 1, m \leq J, m + +$ **do**
9:         **for** $k = n + 1, k \leq M, k + +$ **do**
10:            $R(n) \leftarrow R(n) + x_{m,k} x_{m,k-n}^*$;
11:         **end for**
12:     **end for**
13:     $R(n) \leftarrow R(n)/J(M - n)$;
14:     $\boldsymbol{R}(n) \leftarrow R(n)$;
15: **end for**
16: **for** $n = 1; n \leq N; n + +$ **do**
17:     $G(N) \leftarrow G(N) + n \arg\{\boldsymbol{R}(n)\}$;
18: **end for**
19: $G(N) \leftarrow 6G(N)/N(N + 1)(2N + 1)$;
20: Obtain the CFO estimator $\hat{\varphi}$ by the quantization function $G(N)$, and $\hat{\varphi} \stackrel{\Delta}{=} \hat{\omega} T_c = 2\pi \hat{f} T_c = G(N)$.
21: **return** $\hat{\varphi}$.

## 6. Numerical Results and Discussion

This section exhibits the bit error rate (BER), symbol error rate (SER) and packet error rate (PER) performance of our proposed MSD scheme over a pure AWGN channel and a slow Rayleigh fading channel. The simulation parameters used in this paper are listed in detail in Table 1.

**Table 1.** Parameters used in simulations.

| Parameter | Detailed Description |
|---|---|
| Detection scheme | Multiple-symbol detection |
| Channel condition | Pure AWGN and slow Rayleigh fading |
| Data modulation | Offset QPSK |
| Compensation scheme | Precompensation |
| Timing synchronization | Perfect |
| Power of the complex noise | 1/SNR |
| Symbols | 16-ary quasi-orthogonal |
| Payload length of PSDU (bits) | 176 |
| Spreading factor | 32 |
| Chip rate (M chip/s) | 2 |
| preamble length $J$ | 8 |
| CPO $\theta$ (rads) | Uniform distribution in $(-\pi, \pi)$ |
| Carrier frequency (MHz) | 2480 |
| CFO $f$ (ppm) | Symmetrical triangular distribution in $(-80, 80)$ |
| Computer version | Win10_64 bit |
| Computer central processing unit (CPU) | Intel Core i5-4210U 2.40GHz (Intel, Santa Clara, CA, USA) |

### 6.1. Effect of the Number of Local Metric Factor p on the Detection Performance

For the full MSD scheme, the number of local metric factor $p$ of the decision metric $Z_{j(i-1)+l,p}$ affect the detection performance. In order to observe the specific impact of the local metric factor $p$ on the full MSD scheme, we assume that the CFO is perfectly estimated and compensated (i.e., without CFO). Figure 3 shows the comparisons of the detection performance of the optimal symbol-by-symbol coherent detection (SCD), the optimal symbol-by-symbol noncoherent detection (SND) and the full MSD scheme. We can observe from Figure 3c that when the CFO is perfectly compensated, the PER performance of full MSD scheme is significantly improved compared to the optimal SND scheme; at the PER of $10^{-2}$, the full MSD scheme can achieve a 0.8 dB gain. In particular, when the detection window length is constant, with the increasing of $p$, the detection performance of the full MSD scheme gradually approaches the optimal SCD scheme; when the value of $p$ is fixed, the detection performance of full MSD scheme improves with the detection window length increasing. Hence, as the detection window length and the number of local metric factor $p$ increase at the same time, no significant performance gain is observed here. That is to say, when the factor $p$ is set to be 2, it can meet the need of IEEE 802.1.4g O-QPSK receivers.

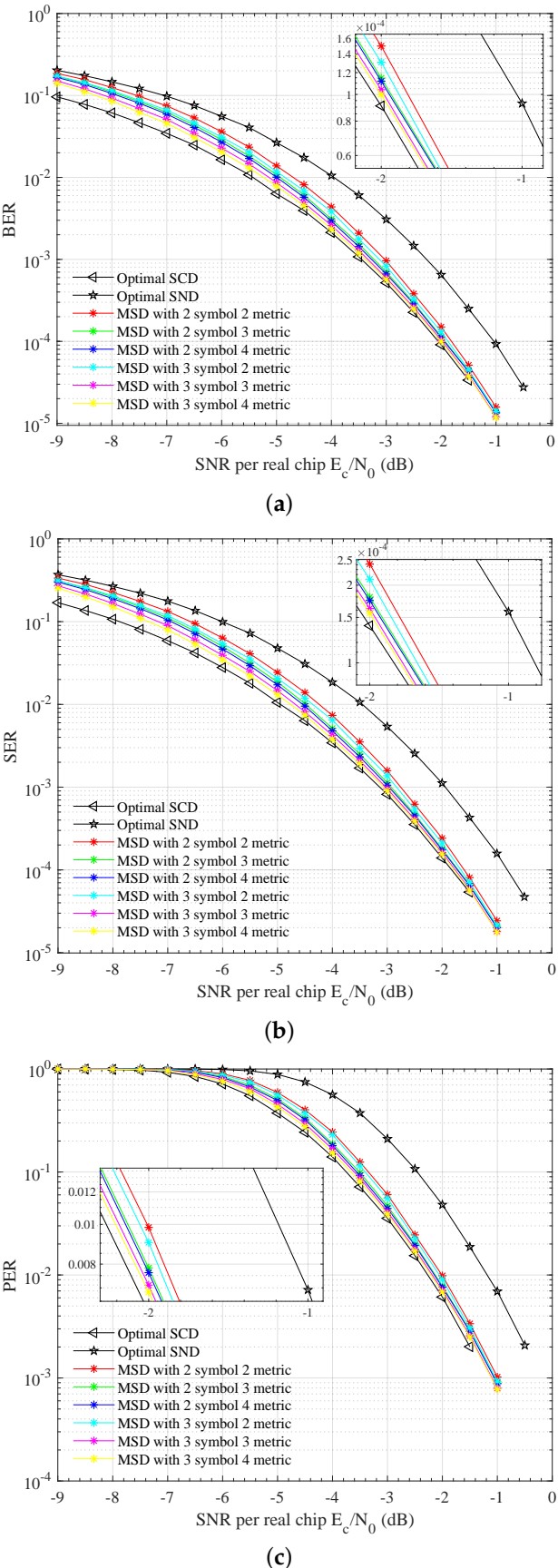

**Figure 3.** Detection performance impact of factor *p* under the proposed MSD scheme over a pure additive white Gaussian noise (AWGN) channel. (**a**) bit error rate (BER) performance; (**b**) symbol error rate (SER) performance; (**c**) packet error rate (PER) performance.

## 6.2. Detection Performance Influence of the Truncated Number N of Differential Chips

The truncated number *N* of the differential chip can effectively improve the performance of our MSD scheme for the IEEE O-QPSK receivers. In this subsection, we research the detection performance comparisons of the proposed MSD scheme versus diverse estimators over a pure AWGN channel under different observation window lengths and different truncated number *N*. Figures 4 and 5 are the PER performance comparisons of our proposed receiver when the observation window length is set to be 2 and 3, respectively. As shown in Figure 4a, we can observe that under a pure AWGN channel, the PER performance of the proposed MSD scheme is improved obviously when truncated number *N* increases from 1 to 3; when N increases from 3 to 5, significant PER performance loss and error floor appears. That is, when the truncated number *N* is set to be 3, the proposed MSD scheme can obtain the optimal PER performance, while the PER performance of the receiver with full estimator and the other two simplified estimators in (18) and (19) are better than optimal SND. In particular, at PER of $1 \times 10^{-2}$ and $N = 3$, compared with the full estimator in (16), our simplified estimator in (19) does not observe significant performance loss; in contrast with the optimal SND, 0.7 dB gains can be achieved. Hence, it can be concluded that the performance requirement of the IEEE 802.15.4g O-QPSK receivers can be met when the truncated number *N* is set to be 3 over the pure AWGN channel. For obtaining the optimal PER performance, in other simulation experiments in this work, we choose the optimal truncated number *N*, that is, $N = 3$.

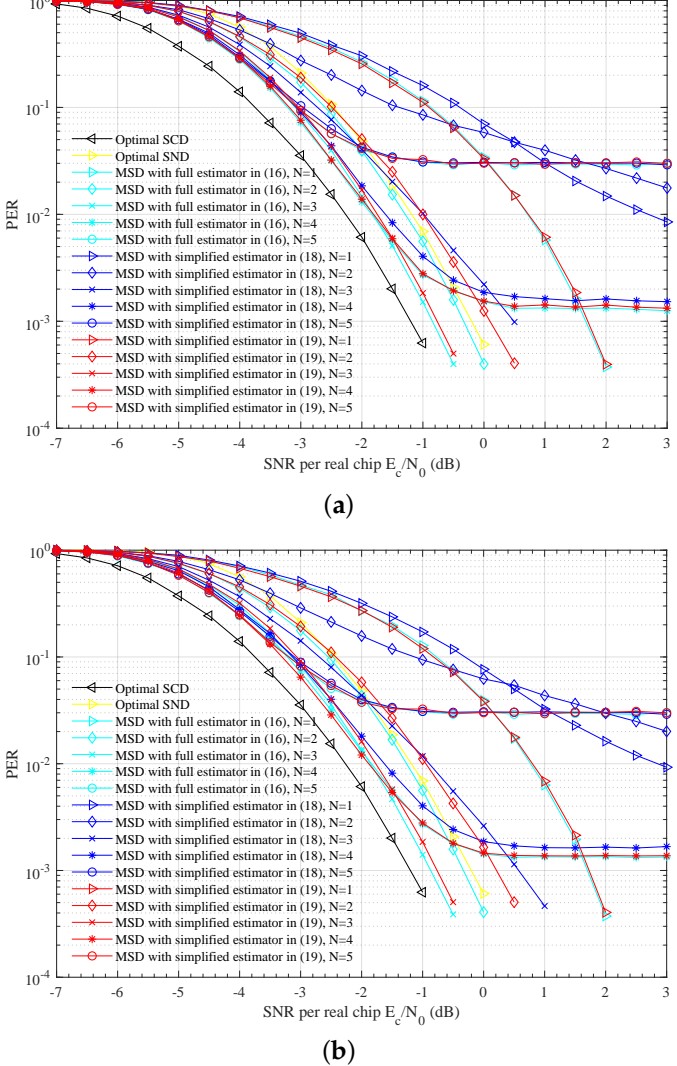

(**a**)

(**b**)

**Figure 4.** *Cont.*

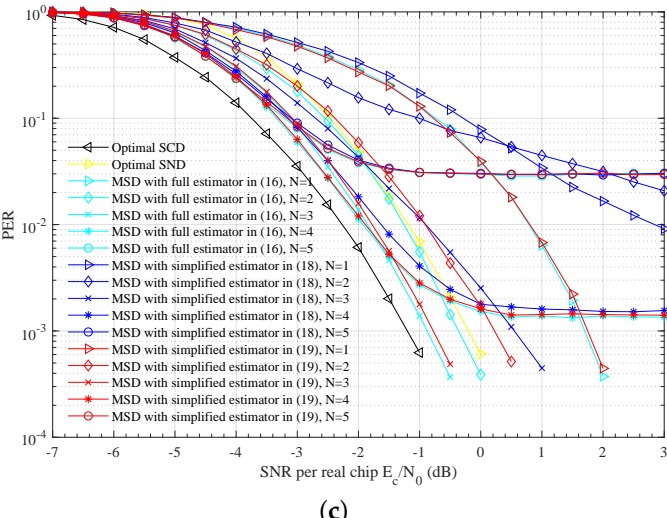

(**c**)

**Figure 4.** Detection performance impact of the truncated number *N* of differential chips under the proposed MSD scheme over a pure AWGN channel when the observation window length *j* = 2. (**a**) PER performance with full estimator in (16); (**b**) PER performance with simplified estimator in (18); (**c**) PER performance with simplified estimator in (19).

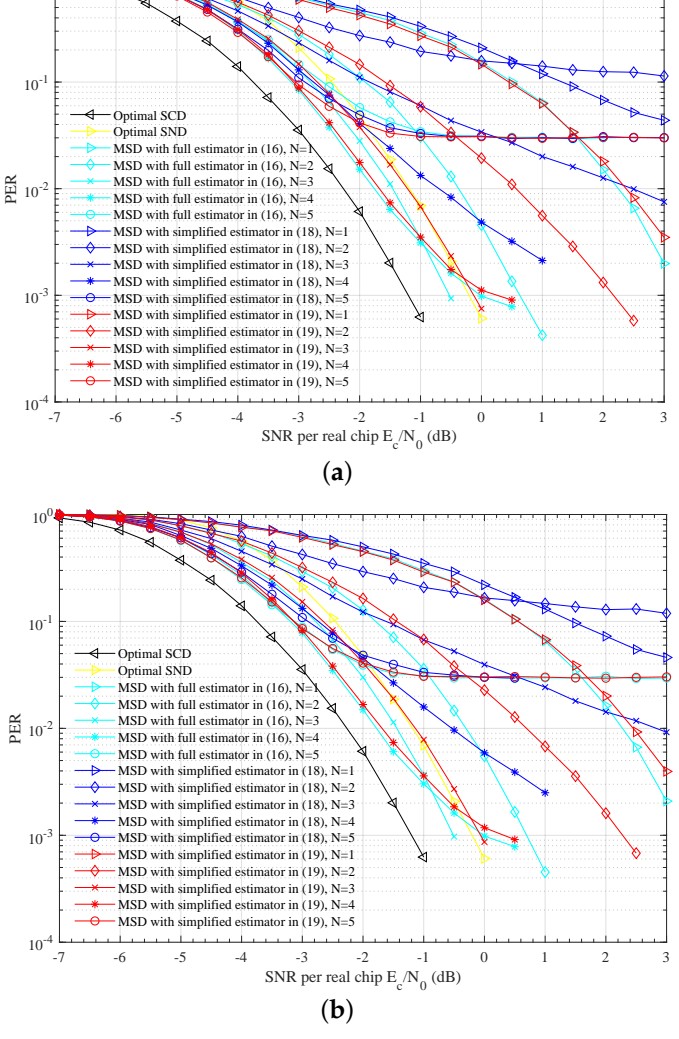

(**a**)

(**b**)

**Figure 5.** *Cont.*

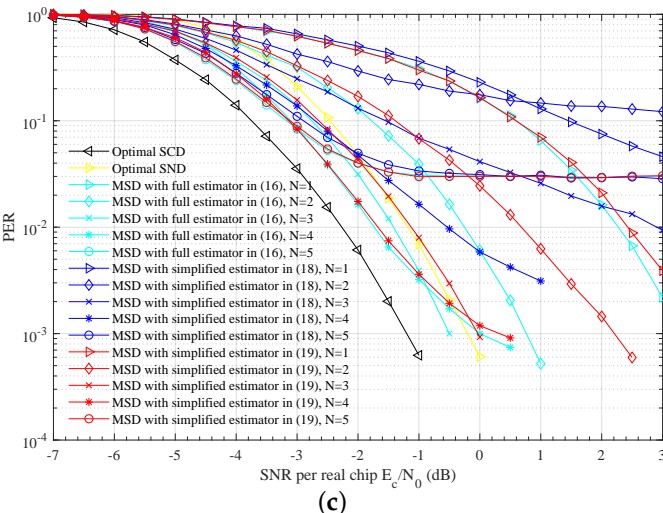

**(c)**

**Figure 5.** Detection performance impact of the truncated number *N* of differential chips under proposed MSD scheme over a pure AWGN channel when the observation window length *j* = 3. (**a**) PER performance with full estimator in (16); (**b**) PER performance with simplified estimator in (18); (**c**) PER performance with simplified estimator in (19).

### 6.3. Detection Performance of the Proposed MSD Scheme over a Pure AWGN Channel

Figure 6 shows the performance comparisons of the proposed MSD scheme under a pure AWGN channel with different observation window lengths and decision metrics, wherein the truncated number *N* is set to be 3 for more reliable results. As shown in Figure 6, when the observation window length is fixed, with the increasing of the number of the decision metric, the detection performance of our enhanced receiver is gradually improved with the full estimator in (16) and the simplified estimator in (19); yet, the receiver performance with the simplified estimator in (18) has not been significantly improved, which is mainly due to the sizeable absolute error produced by the simplified CFOE estimator in (18). In particular, when the number of decision metric is constant, with the observation window length increasing, there is no significant performance gain, which is also caused by the absolute error. Moreover, as shown in Figure 6c, when the observation window length and the number of metric values both are set to be 2, at PER of $1 \times 10^{-2}$, compared to the optimal SND, our enhanced receiver with the simplified estimator in (19) can achieve 0.6 dB gains. In other words, adopting the simplified estimator (19), we can significantly reduce the complexity of the CFO estimator, while ensuring the performance gain. Therefore, our simplified CFO estimator is compatible with the simplified MSD scheme proposed in this paper.

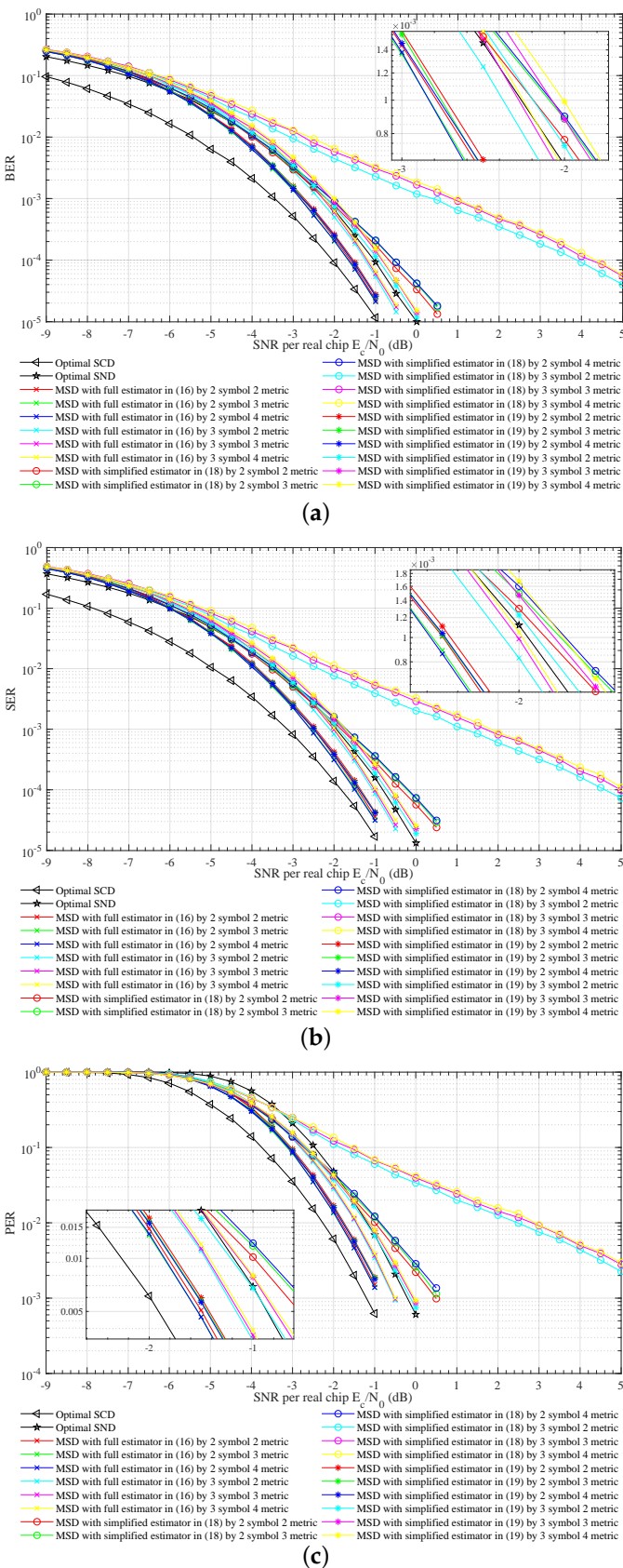

**Figure 6.** Detection performance comparisons of various detection schemes over a pure AWGN channel. (**a**) BER performance; (**b**) SER performance; (**c**) PER performance.

### 6.4. Detection Performance of the Proposed MSD Scheme over a Slow Rayleigh Fading Channel

The BER, SER and PER performance of the MSD and SBSD schemes is shown in Figure 7, where the channel model is a slow Rayleigh fading channel, and the truncated number *N* of the differential chip is set to be 3. We can observe from Figure 7c that our MSD scheme performs well compared with the SBSD scheme. At the PER of $1 \times 10^{-2}$, the two-symbol detection scheme with full estimator and 3 metrics can achieve 3.8 dB gains. Moreover, for the two-symbol detection scheme with the full estimator, when the number of decision metric is increased from 2 to 4, there is no intuitive performance gain; that is, the O-QPSK receivers can obtain satisfactory detection performance when the number of decision metric is set to be 2. We can also observe that when the observation window is increased from 2 to 3 symbol periods, the detection performance is exceptionally similar, and 2 symbol observation windows are sufficient to meet the performance requirements of IEEE 802.15.4g O-QPSK receivers. In particular, for the two-symbol detection scheme using 2 decision metrics, compared with the full estimator in (16), the detection performance loss of the approximate simplified estimator in (18) and (19) is not apparent, which is also the expected result in the design of low-power WSNs nodes.

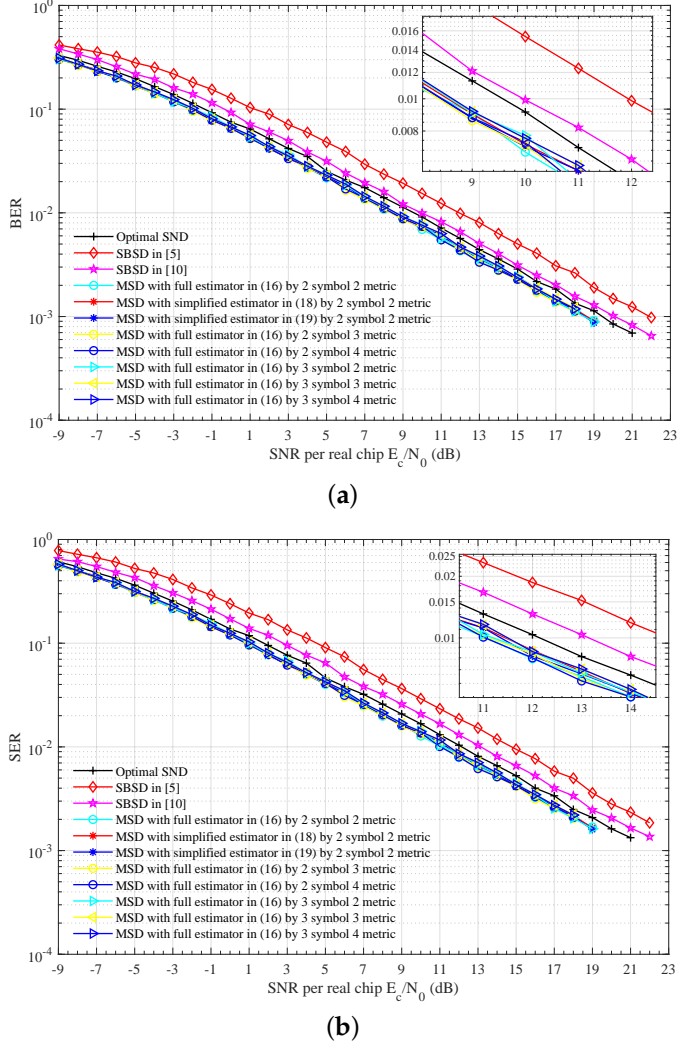

(**a**)

(**b**)

**Figure 7.** *Cont.*

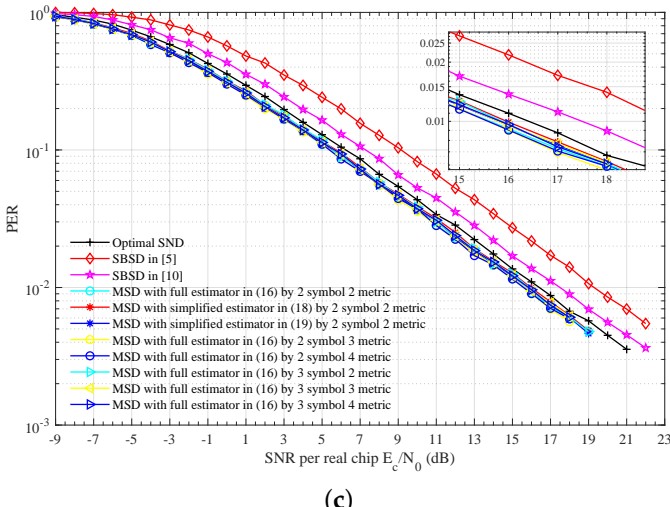

(**c**)

**Figure 7.** Detection performance comparisons of various detection schemes over a slow Rayleigh fading channel. (**a**) BER performance; (**b**) SER performance; (**c**) PER performance.

*6.5. Transmitter Energy Consumption for the Proposed MSD Scheme*

This subsection evaluates the O-QPSK receiver with the proposed MSD scheme in Atmel AT86RF215 [31,32]. Specifically, we observed the transmission energy consumption of the proposed MSD scheme configured various estimators and numbers of metric values when the observation window length $j$ is set to be 2. The transmission consumed by the receiver is $E = IVN_p/f_{tx}$ [33]. Here, $I$ is the transmission current, the supply voltage $V$ assumed to be $3V$, $N_p$ is the chip length of the physical layer protocol data unit (PPDU), and the chip transmission rate $f_{tx}$ is $2M/chip$. Tables 2 and 3 respectively show the transmission supply current and the transmission energy gain of the receiving node under the pure AWGN and slow Rayleigh noise channels when the proposed MSD scheme is configured with various estimations.

The transmission current $I$ consumed by the transmit-only nodes depends on the distributed transmission energy of each chip symbol. As shown in Table 3, compared with the SBSD scheme, when $PER = 1 \times 10^{-2}$, the proposed MSD scheme configured full estimation in (16) can achieve 0.63–0.67 dB gains in the distributed transmission energy, while 2.71%–2.88% of transmission energy can be saved under the pure AWGN channel; the proposed MSD scheme configured simplified estimator in (19) reduces the complexity of the full MSD scheme while saving 2.40%–2.53% of transmission energy. We can observe from Table 3 that the required transmission energy per chip is also reduced by 0.45–0.83 dB, while saving 1.94%–3.56% more energy than the SBSD scheme under the slow Rayleigh fading channel at $PER = 1 \times 10^{-2}$.

**Table 2.** At $PER = 1 \times 10^{-2}$, supply current for transmission and transmission energy gain achieved by the proposed MSD scheme in the sink node over the pure AWGN channel.

| The Proposed MSD Scheme with $j = 2$ | SNR Gain (dB) | Supply Current $I$ (mA) | Consumed Energy (µJ) | Gain (µJ) | Energy Saving (%) |
|---|---|---|---|---|---|
| full estimator in (16) by 4 metric | 0.67 | 25.12 | 62.69 | 1.86 | 2.88 |
| full estimator in (16) by 3 metric | 0.66 | 25.13 | 62.72 | 1.83 | 2.84 |
| full estimator in (16) by 2 metric | 0.63 | 25.16 | 62.80 | 1.75 | 2.71 |
| simplified estimator in (19) by 4 metric | 0.59 | 25.21 | 62.92 | 1.63 | 2.53 |
| simplified estimator in (19) by 3 metric | 0.58 | 25.22 | 62.95 | 1.60 | 2.48 |
| simplified estimator in (19) by 2 metric | 0.56 | 25.24 | 63.00 | 1.55 | 2.40 |

**Table 3.** At $PER = 1 \times 10^{-2}$, supply current for transmission and transmission energy gain achieved by the proposed MSD scheme in the sink node over the normalized slow Rayleigh fading channel.

| The Proposed MSD Scheme with $j = 2$ | SNR Gain (dB) | Supply Current $I$ (mA) | Consumed Energy (μJ) | Gain (μJ) | Energy Saving (%) |
|---|---|---|---|---|---|
| full estimator in (16) by 4 metric | 0.83 | 24.94 | 62.25 | 2.30 | 3.56 |
| full estimator in (16) by 3 metric | 0.82 | 24.95 | 62.27 | 2.28 | 3.53 |
| full estimator in (16) by 2 metric | 0.78 | 24.99 | 62.38 | 2.17 | 3.36 |
| simplified estimator in (19) by 2 metric | 0.61 | 25.18 | 62.85 | 1.70 | 2.63 |
| simplified estimator in (18) by 2 metric | 0.45 | 25.36 | 63.30 | 1.25 | 1.94 |

### 6.6. Robustness of the Proposed MSD Scheme to CPO over Pure AWGN Channel

This subsection investigates the PER performance of our proposed MSD scheme versus dynamic CPOs in a pure AWDN channel, wherein the observation window length $j$ is fixed as 2. Figure 8a–c are the PER performance comparisons of our simplified MSD scheme with dynamic CPOs under diverse CFO estimators, respectively. The distribution of phase $\theta$ obeys the Wiener process and can be modeled as $\theta_{m+1} = \theta_m + \Delta m$, where $\Delta m$ is a zero-mean Gaussian random variable with known variance $\sigma_m^2$. The initial phase $\theta_1$ obeys the uniform distribution on $(-\pi, \pi)$. As shown in Figure 8a, our simplified receiver is robust to phase jitter, and there is no significant performance degradation when the standard deviation $\sigma_m$ of phase jitter increases to 3 degrees. In particular, it can be seen that different estimators all show irreducible error levels with the SNR increasing.

Further, Figure 9 exhibits the BER, SER and PER performance comparisons of the proposed MSD scheme with different estimators under dynamic CPOs. We can observe from Figure 9 that the proposed MSD scheme with the full estimator in (16) performs well under dynamic CPOs, while the other two simplified estimators in (18) and (19) both show acceptable performance degradation. Specially, with the decreasing of standard deviation $\sigma_m$, the influence of $e^{jm}$ on $Y[i]$ decreases gradually, which is mainly caused by the random phase increment $e^{jm}$ generating a phasor in the output $Y[i]$.

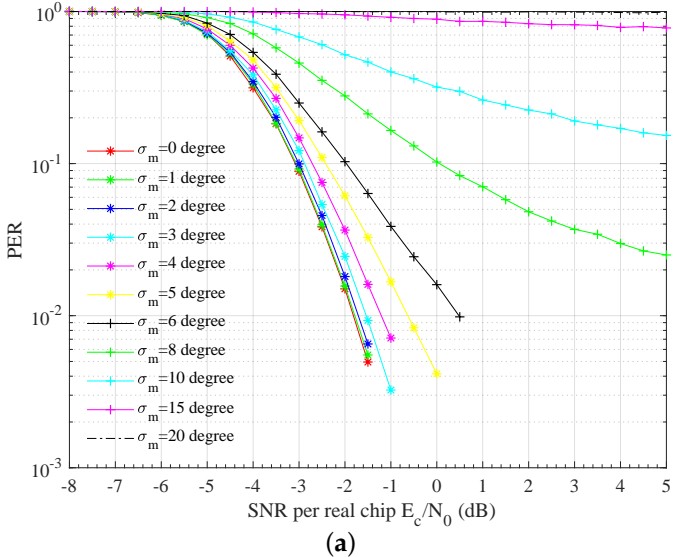

(**a**)

**Figure 8.** *Cont.*

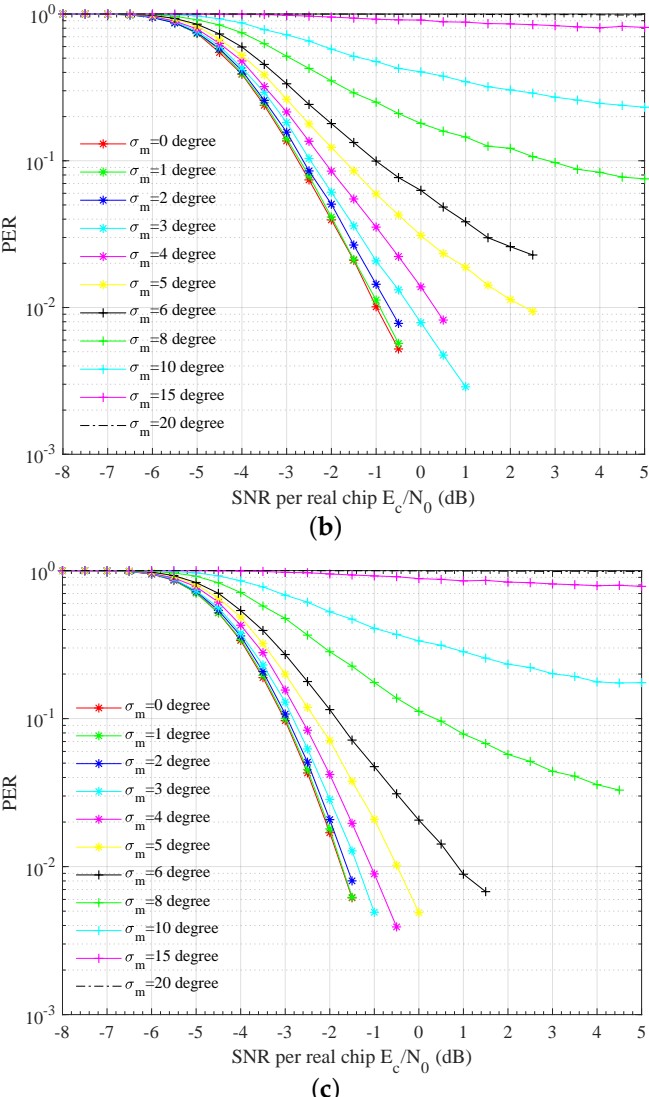

**Figure 8.** Detection performance comparisons of our proposed scheme versus carrier frequency offsets (CFOs) over a pure AWGN channel. (**a**) PER performance with full estimator in (16); (**b**) PER performance with simplified estimator in (18); (**c**) PER performance with simplified estimator in (19).

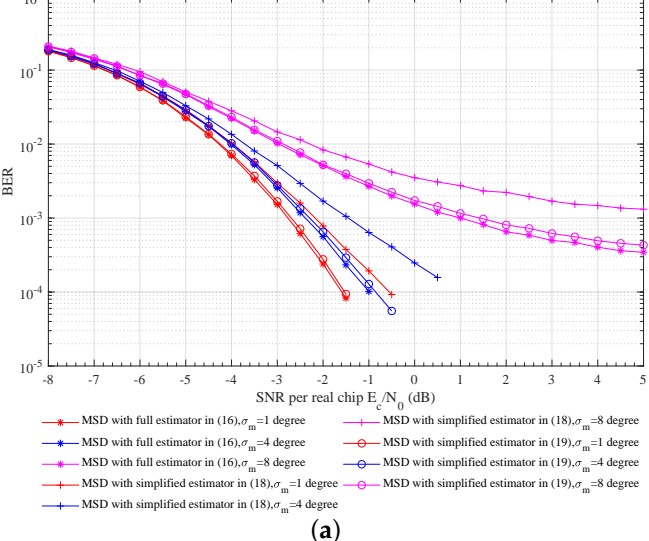

**Figure 9.** *Cont.*

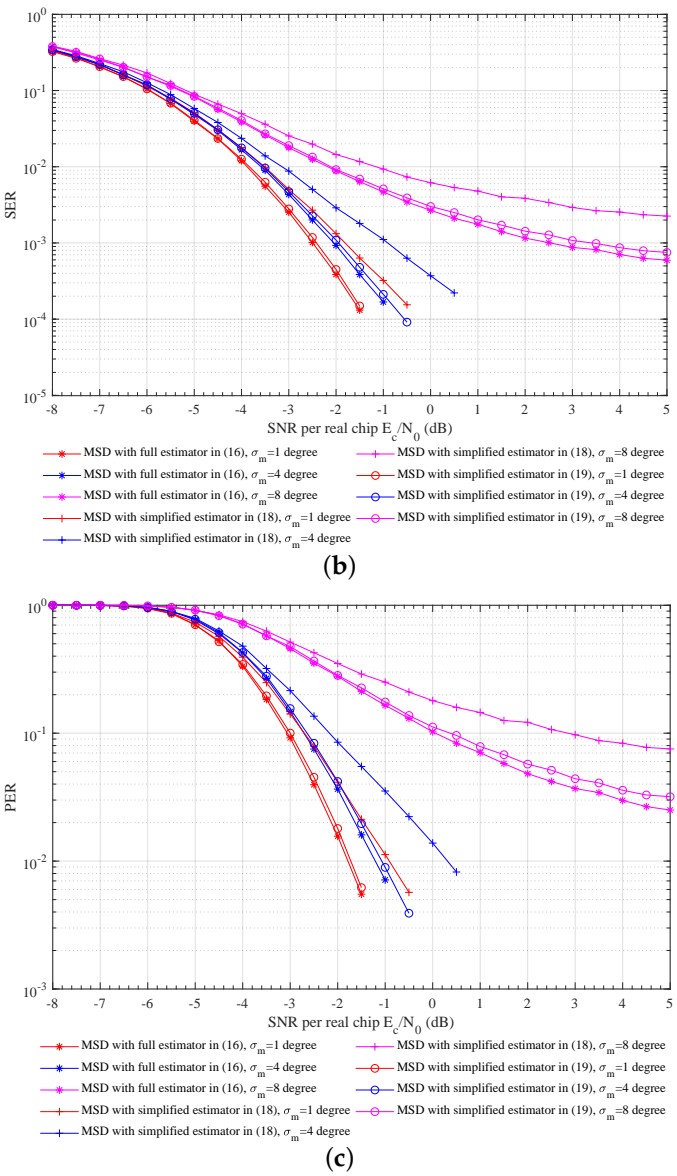

**Figure 9.** Detection performance comparisons of our proposed scheme under various estimators versus carrier phase offset (CPO) over a pure AWGN channel. (**a**) BER performance; (**b**) SER performance; (**c**) PER performance.

*6.7. Robustness of the Proposed MSD Scheme to CFO over Pure AWGN Channel*

As shown in Figure 10, we set both the observation window length *j* and the local metric factor *p* to 2 and evaluate the BER, SER and PER performance of the simplified MSD scheme with dynamic CFOs over a pure AWGN channel. In particular, full estimators in (16) and (17) provide a benchmark for comparison. From Figure 10, we can observe that our simplified estimator performs well when CFOE is in the range of $(-20, 20)$ ppm; when the CFOE is close to $\pm 30$ ppm and $\pm 50$ ppm, the performance of our simplified estimator suffers to a certain extent, and this loss gradually increases with the increasing of the SNR, which is primarily caused by the absolute estimation error gradually approaching the maximum value. Obviously, compared to the simplified estimator in (18) involving $\tan^{-1}(x) = x$, the simplified estimator in (19) involving $\sin^{-1}(x) = x$ is more robust to dynamic CFOs. Yet, under the condition of only considering CPO (i.e., without CFO), our MSD scheme provides much less revenue for all CFOs. Hence, the proposed MSD scheme with the simplified estimator in (19) is insensitive to CFO.

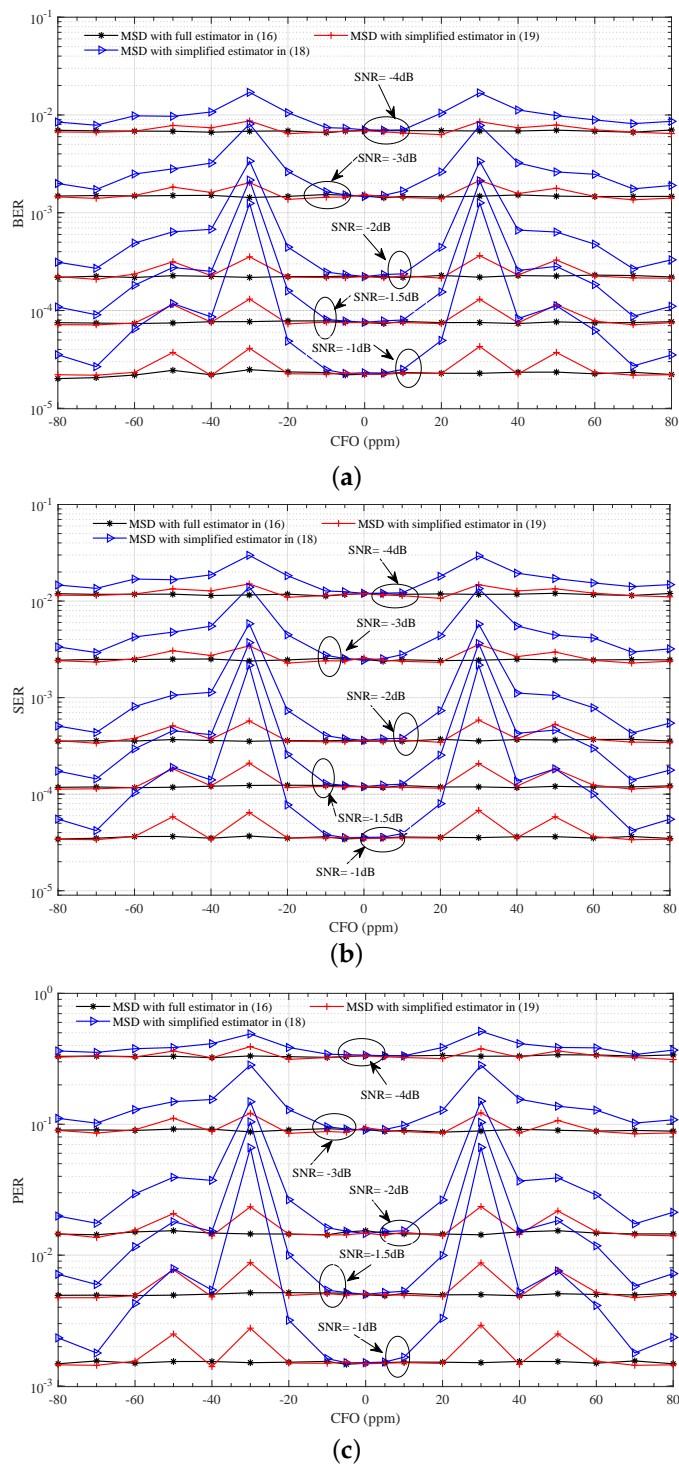

**Figure 10.** Detection performance comparisons of our proposed scheme under various estimators with dynamic CFOs over a pure AWGN channel. (**a**) BER performance; (**b**) SER performance; (**c**) PER performance.

*6.8. Complexity Analysis of the Proposed MSD Scheme*

In this subsection, the algorithm complexity comparisons between the proposed MSD scheme and the full MSD scheme are investigated. We notice that form Table 4 the number of $Y$ to be searched in each observation block is 8 for our proposed MSD scheme while it is 256 in the full MSD scheme. In particular, when the observation window length $j$ is set to be 2, the search number of $Y[i]$ for our

proposed scheme is reduced by 64 times; when *j* is set to be 3, it is reduced by 512 times, which is consistent with our analysis.

**Table 4.** Complexity comparison.

| Number | Number of $\hat{Y}[i]$ to be Search on Proposed MSD Scheme ($16^j$) | Number of $\hat{Y}[i]$ to be Search on Full MSD Scheme ($2^j$) | Reduction Factor ($8^j$) |
|---|---|---|---|
| Observation window length $j$ = 2 | 256 | 4 | 64 |
| Observation window length $j$ = 3 | 4096 | 8 | 512 |

Furthermore, we investigate the complexity of our proposed MSD scheme from the aspect of the average running time. Specifically, when the observation window length is fixed as 2 and the local metric factor *p* is also set to be 2, we contrast the average running time of $10^5$ data packets of different detection schemes. As shown in Figure 11, at the SNR of −2 dB, the average packet running time of the proposed MSD scheme with full estimator is $1.383 \times 10^{-2}$ s, and the SBSD scheme in [5] is $3.278 \times 10^{-2}$ s, which is 2.37 times that of the former; the SBSD scheme in [10] is $1.828 \times 10^{-2}$ s, which is 1.32 times of the proposed MSD scheme. Generally, the implementation complexity of the full MSD scheme is much higher than that of the SBSD scheme. Yet, as shown in Figure 11, compared with the other two SBSD schemes in [5,10], the average packet running time of our proposed MSD scheme is observably less, which is what is expected in the design of ultra-low-power wireless communication systems.

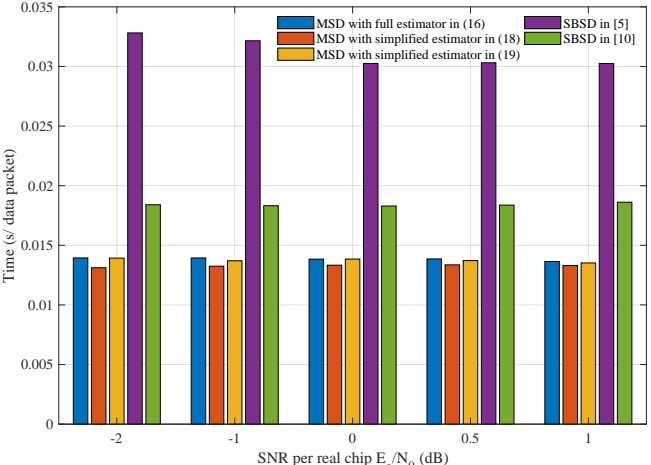

**Figure 11.** Comparisons of average running time per data packet under various detection schemes over a pure AWGN channel.

## 7. Conclusions and Future Work

This paper introduces a novel and simplified MSD scheme over slow fading channel to reduce the high complexity of the full MSD scheme for IEEE 802.15.4g O-QPSK receivers. The simulation results indicate that our proposed MSD scheme can decrease the number of the decision metric for the full MSD scheme from $16^j$ to $2^j$, while the performance loss is acceptable. Moreover, our simplified CFOE estimation strategy can effectively compensate for the impact of residual CFO, which is also appropriate for the full MSD scheme. In addition, the CFOE quantization function can save a lot of transmitting energy of only-transmit nodes no matter which estimator is configured, which is pretty attractive for low-power, low-cost IEEE 802.15.4g O-QPSK receivers.

In this work, although the slow fading channel has been considered, the proposed MSD scheme can also be applicable to the fading channel under fast-changing carrier phase and carrier frequency. Simulation, hardware implementation and software testing are equally critical for new algorithm, and software defined radio (SDR) is a good platform that is worth considering for algorithm testing

in true environment. Furthermore, there are some future research directions. Orthogonal frequency division multiplexing (OFDM) technology supports higher data transmission rate [34–37]. Channel coding, especially low density parity check (LDPC) code, can effectively improve transmission reliability [38]. In addition, our current research focuses on noncoherent detection of single-carrier modulation, and the proposed MSD scheme cannot be straightway applied to multicarrier systems.

**Author Contributions:** Conceptualization, C.S., G.Z. and H.L.; data curation, C.S.; formal analysis, C.S. and G.Z.; funding acquisition, G.Z., C.H., L.W. and D.W.; investigation, C.S., J.T. and H.W.; methodology, C.S. and G.Z.; project administration, G.Z., J.T. and L.W.; software, C.S.; validation, C.S.; writing—original draft, C.S.; writing—review and editing, C.S. and G.Z. All authors have read and agreed to the published version of the manuscript.

**Funding:** This work was partially supported by the National Natural Science Foundation of China (61701172, 41605122, 61701059, 61701062, 61801170, 61801171, and 61772175), the Scientific and Technological Innovation Team of Colleges and Universities in Henan Province (20IRTSTHN018), Open Foundation of the Key Laboratory of Middle Atmosphere and Global Environment Observation, Institute of Atmospheric Physics, Chinese Academy of Sciences, Postdoctoral Science Foundation of University of Electronic Science and Technology of China (Y02006023601721), the Program for Everest Scholar Talents Development in Tibet University, and the Program for Science & Technology Innovation Talents in the University of Henan Province (Educational Committee) (17HASTIT025).

**Conflicts of Interest:** The authors declare no conflict of interest.

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
