# Peer review of "Reduced-Complexity Multiple-Symbol Detection of O-QPSK Signals in Smart Metering Utility Networks"

_electronics, doi:10.3390/electronics9122049_

Round 1

Reviewer 1 Report

This paper introduces a multiple-symbol detection method for the IEEE 802.15.4g O-QPSK receivers. The proposed low-complexity detection performs close to the optimal detection for a specific set of parameters. The paper is well written, and the work technically sounds and is of interest to the journal readers and the research community at large.

Page 2, Line 28-29: brings difficulties considerably to the detection process. à brings considerable challenges to the detection process.

Please define CFO, CFOE, CPO, PN, and AWGN at their first appearances.

Page 4, Line 109: Specially, à Specifically

Page 4, Line 113: where M is set to be the maximum 16 à where M is set to be the maximum of 16

Page 4, Line 123: the decision statistics here exist a total of 4096 results à the decision statistics contain 4096 values.

Page 4, Line 124: scheme reduces the decision statistic by 512 times à reduces the decision statistics by a factor of 512.

Please provide references for equations (16) and (17) if they were used in the literature. I also suggest that authors should provide rationales behind their choice of these functions.

6.1 Detection Performance Influence of the Number of Local Metric Factor p à 6.1 Effect of the Number of Local Metric Factor, p, on the Detection Performance

Please check the paper for minor grammatical issues.

Author Response

Dear reviewer,

Thank you very much for your comments on our manuscript. According to your comments and suggestions, we have further revised and improved the manuscript. Please see the attachments for the specific reply and the revised manuscript.

We are uploading (a) our point-by-point response to the comments (below) (Response to Reviewers 1 Comments), (b) an updated manuscript with yellow highlighting indicating changes (Supplemental File for Review), and (c) a clean updated manuscript without highlights (PDF Main Document).

Best regards,

All authors.

Reviewer 2 Report

The paper is reasonably well written and it states clearly its main contributions. 

Since many references are more than a decade ago, it would be interesting for many readers to have some explanation why in this field progress is relatively slow.

Author Response

Dear reviewer,

Thank you very much for your comments on our manuscript. According to your comments and suggestions, we have further revised and improved the manuscript. Please see the attachments for the specific reply and the revised manuscript.

We are uploading (a) our point-by-point response to the comments (below) (Response to Reviewers 2 Comments), (b) an updated manuscript with yellow highlighting indicating changes (Supplemental File for Review), and (c) a clean updated manuscript without highlights (PDF Main Document).

Best regards,

All authors.

Reviewer 3 Report

Authors represented algorithm which reduce multiple-symbol detection of O-QPSK modulation. They claim that developed algorithm outperform existing one in computation time. In paper are many different simulation scenarios represented with graphs it is somehow hard to follow the paper. However, I have some questions and comments:

  1. Paper is focused on low power wireless sensor networks and then you used Intel i5 processor for decoding symbols. Did you try implement algorithm using low power and low performance microcontroller?
  2. You performed only simulation. Did you implement algorithm on one of many software defined radios? I think SDR is good platform for test algorithm in true environment.

Simulations are crucial and useful but never consider all parameters from real word. So, I really missing implementation of algorithm on real hardware and in real environment. At least implementation on software defined radio could improve the quality of paper.

Author Response

Dear reviewer,

Thank you very much for your comments on our manuscript. According to your comments and suggestions, we have further revised and improved the manuscript. Please see the attachment for the specific reply and the revised manuscript.

We are uploading (a) our point-by-point response to the comments (below) (Response to Reviewers 3 Comments), (b) an updated manuscript with yellow highlighting indicating changes (Supplemental File for Review), and (c) a clean updated manuscript without highlights (PDF Main Document).

Best regards,

All authors.

Round 2

Reviewer 3 Report

Thanks for comments and answers. I'm satisfied with answers.